# Anti-Inflammatory Drug Therapy in Chronic Subdural Hematoma: A Systematic Review and Meta-Analysis of Prospective Randomized, Double-Blind and Placebo-Controlled Trials

**DOI:** 10.3390/ijms232416198

**Published:** 2022-12-19

**Authors:** Martin Vychopen, Erdem Güresir, Johannes Wach

**Affiliations:** Department of Neurosurgery, University Hospital Leipzig, 04103 Leipzig, Germany

**Keywords:** inflammation, anti-inflammatory drug therapy, chronic subdural hematoma, mortality, meta-analysis

## Abstract

Althoughanti-inflammatory drug therapy has been identified as potentially beneficial for patients suffering from chronic subdural hematoma (cSDH), contemporary literature presents contradictory results. In this meta-analysis, we aimed to investigate the impact of anti-inflammatory drug therapy on mortality and outcome. We searched for eligible randomized, placebo-controlled prospective trials (RTCs) on PubMed, Embase and Medline until July 2022. From 97 initially identified articles, five RTCs met the criteria and were included in our meta-analysis. Our results illustrate significantly lower rates of recurrent cSDH (OR: 0.35; 95% CI: 0.21–0.58, *p* = 0.0001) in patients undergoing anti-inflammatory therapy. In the subgroup of patients undergoing primary conservative treatment, anti-inflammatory therapy was associated with lower rates of “switch to surgery” cases (OR: 0.30; 95% CI: 0.14–0.63, *p* = 0.002). Despite these findings, anti-inflammatory drugs seemed to be associated with higher mortality rates in patients undergoing surgery (OR: 1.76; 95% CI: 1.03–3.01, *p* = 0.04), although in the case of primary conservative treatment, no effect on mortality has been observed (OR: 2.45; 95% CI: 0.35–17.15, *p* = 0.37). Further multicentric prospective randomized trials are needed to evaluate anti-inflammatory drugs as potentially suitable therapy for asymptomatic patients with cSDH to avoid the necessity of surgical hematoma evacuation on what are predominantly elderly, vulnerable, patients.

## 1. Introduction

Chronic subdural hematoma (cSDH) is a commonly diagnosed neurological disorder among elderly patients. Due to global population aging and spreading of slice imaging, the overall incidence increased significantly, from 8.2 to 48 per 100,000/year [1]. Against this backdrop, there is an increasing demand to investigate sufficient novel adjuvant therapy methods to surgery or alternative methods to surgery for asymptomatic patients. To date, surgical evacuation is considered the main avenue of medical treatment [2,3,4,5,6,7].

Traumatic injury, and corresponding intracranial hemorrhage, are the key sources of development of cSDH. However, sustained inflammatory burden is suggested as one of the major drivers in cSDH fluid collection and membrane growth [8]. Vascular endothelial growth factor (VEGF) in the cSDH fluid collection is increasingly discussed and investigations suggest that it may be secreted by inflammatory cellular components, such as neutrophils, within the fluid collection, and the endothelial cells of the external membrane, or by macrophages infiltrating the cSDH [9,10,11]. Furthermore, the concentration of interleukin-6 (IL-6) in the cSDH fluid has been found to be significantly increased in patients with recurrent cSDH, compared to those without a recurrence [12].

Previous clinical trials investigated several anti-inflammatory (e.g., corticosteroids, atorvastatin, celecoxib) therapy options to optimize the clinical care of those predominantly elderly and vulnerable cSDH patients. For instance, corticosteroids are known for their anti-inflammatory functions, which alter the gene expression profile, the transcription of mediators of inflammatory responses, such as cytokines or chemokines, and the polarization of macrophages [13,14,15,16]. Moreover, statins can accelerate hematoma resorption and reduce brain edema by enhancing Treg cells in the brain, which act as negative regulators of inflammation [17]. To date, there is no general consensus for a standardized prescription of an anti-inflammatory drug for cSDH patients.

The present meta-analysis aimed to investigate the existing evidence and identify effective anti-inflammatory interventions for cSDH patients with regard to neurological outcome, mortality, and cSDH growth. 

## 2. Methods

In this meta-analysis, the authors strictly followed the PRISMA checklist (see Appendix A) [18] and the Cochrane Handbook for systematic Reviews of Interventions Version 6.3 [19]. The systematic review and meta-analysis were registered in POSPERO prior on completion of the initial search.

### 2.1. Inclusion and Exclusion Criteria

The authors conducted a systematic search of the Pubmed, Embase and Medline databases in July 2022, for the term “chronic subdural hematoma”. The search was limited to “randomized controlled trials”, “human studies”, “clinical trials” and “English”. The literature search included all results until 30 June 2022. The inclusion criteria were formulated according to the PICOS (population, intervention, comparator, outcomes and study design) framework [20]. These criteria were as follows: patients had undergone treatment for chronic subdural hematoma; relevant anti-inflammatory drug therapies were performed; results were compared to a placebo control; all results of the prespecified endpoints were reported; and the trials were defined as prospective randomized, placebo-controlled, and double-blinded studies. The following types of records were excluded: reviews, study protocols, letters, conference abstracts, unpublished papers, animal experiments, and trials with insufficient data (e.g., randomized controlled trials without a placebo control or that were not double-blinded in setting). Furthermore, previous meta-analyses and reviews were searched for studies matching our inclusion and exclusion criteria. 

The identified articles were further examined in a stepwise workflow: (1) titles of the studies, (2) abstract of the study and, finally, (3) the two authors screened the full texts independently (MV and JV) until all retrieved studies were either included or excluded. Any disagreement between the two authors was settled by the third author (EG).

### 2.2. Types of Studies

To conduct the meta-analysis, we included randomized, double-blind, placebo-controlled clinical trials evaluating the use of anti-inflammatory drugs regarding long-term neurological outcome, mortality, and the need for secondary surgery due to recurrence/hematoma expansion in patients with chronic subdural hematoma. 

Modified Ranking scale was used to measure the neurological outcome, which was, subsequently, evaluated in accordance with the definition of poor outcome as a score of 3–6 and good outcome as a score of 0–2 [21]. The Markwalder grading score was dichotomized into good (0–1) and poor (2–4) outcome.

To evaluate the effect of anti-inflammatory therapy on recurrence rate/hematoma expansion, relevant patients were only those with the need for secondary surgery. We defined secondary surgery as either failure of conservative treatment and subsequent switch to surgery, or necessity for second surgery after primary hematoma evacuation. Symptom-free patients with radiological signs of subdural hematoma expansion were not included in the analysis of recurrence rate/hematoma expansion if secondary surgery had not been performed [22].

Corticosteroid therapy regimens were investigated regarding the administered dosage. The cumulative dosages of the trials were divided in low dose and high dose corticosteroid therapies. The threshold for the dichotomization of corticosteroid therapies was the median of cumulative corticosteroid dosage of all studies (190 mg). 

### 2.3. Data Extraction and Quality Evaluation

Study names, first authors, year of publication, country, number of centers (mono-, bi- or multi-centric), basic trial design (randomization, allocation, lack of data, lack of reported outcome, intention-to-treat analysis), and other relevant data were extracted as baseline data. The Cochrane Bias Risk Tool was used to investigate the risk of bias (ROB) in the included trials using the software Review Manager Web (RevMan Web Version 5.4.1 from The Cochrane Collaboration, available at revman.cochrane.org (accessed on 31 August 2022)). The following six characteristics regarding risk of bias assessment were included in the analysis: selection bias, performance bias, detection bias, attrition bias, reporting bias, and other sources of bias. Afterwards, a risk of bias summary chart and plot were created.

### 2.4. Statistics

Review Manager Web (RevMan Web Version 5.4.1 from The Cochrane Collaboration) was used to conduct the meta-analyses. To investigate the statistical heterogeneity and inconsistency, x^2^ and I^2^ statistics were used, respectively; an I^2^ value of 50% or more represented substantial heterogeneity. Weight of the relative contribution of the individual studies, based on the sample size, was considered with regard to the estimation of treatment effects. The following three methods were used to assess the publication bias: (1) Funnel plots were created to visually examine the publication bias of included studies; (2) An Egger regression test was used to statistically investigate the funnel plot symmetry. The likelihood of publication bias was determined using the Egger regression intercept two-tailed test and a 5% significance threshold was set [23]; (3) Begg’s test was performed to evaluate the asymmetry of the data [24]. Egger’s and Begg´s tests were performed using MedCalc (Version 20.123 for Windows). Effect sizes were expressed as pooled OR estimates. The following endpoints were investigated using this statistical stepwise workflow: mortality, recurrence, and neurological outcome. 

### 2.5. Grading of the Evidence

The overall certainty of all trials was rated using the Grading of Recommendations, Assessment, Development, and Evaluation (GRADE) approach [25,26]. Criteria to judge included trial limitations (according to Newcastle Ottawa Scale [27]), inconsistency (significant study heterogeneity, I^2^ > 50%), indirectness (characteristics limiting the generalizability of the findings), imprecision (the 95% CI for effect estimates crosses for an essential difference of 5% from the line of unity), and publication bias (small trial effects causing significant evidence).

## 3. Results and Discussion

### 3.1. Literature Search

According to the defined search algorithm (presented in Section 2.1, Section 2.2 and Section 2.3), a total of 97 articles (Figure 1) were identified. After review of the titles, abstracts, and full texts, 92 articles were excluded. Finally, 5 articles, involving 1168 patients, were eligible for the meta-analysis. 

### 3.2. Characteristics of Included Studies

The included studies of the present meta-analysis were published between 2015 and 2021. The summary of major key characteristics of all included trials is provided in Table 1. For further information on included studies, see Table 1. The excluded randomized anti-inflammatory studies are summarized in Appendix A.

The subjects of each individual study were treated for cSDH and were randomized into either anti-inflammatory therapy or placebo therapy. Two studies investigated primarily conservatively treated patients [31,32], one study investigated both surgically or conservatively treated patients [29], and two studies reported only patients who underwent surgical treatment [28,30]. As far as clinical endpoints are concerned, one study reported only on mortality without further data on neurological outcome [32]. Four studies reported on neurological outcome, using the Markwalder grading score (MGS) or the modified Rankin scale [28,29,30,31]. All studies reported on the rates of secondary surgery for recurrence in primarily surgically treated patients or because of switch of therapy in primarily conservatively treated patients.

The duration of drug treatment varied from 14 to 56 days among the included trials. Further anti-inflammatory prospective studies in cSDH [33,34,35,36,37,38,39,40,41,42], which did not fulfill our inclusion criteria, due to having a single-blinded study design, unknown blinding method or the absence of placebo-controlled arms, are summarized in Appendix A.

### 3.3. Risk of Bias Quality Assessment 

All included trials described their methods of randomization and the corresponding allocation system. Double-blinded personnel and patients were present in all trials. Two trials might have an attrition bias. Prespecified outcome was reported in all trials. Overall, no prospective, randomized, double-blind, and placebo-controlled trial showed characteristics indicating a high risk of bias. The frequency of the individual bias of each trial and the overall bias assessment are summarized in Figure 2. The quality assessment protocol detailing the risk of bias and author judgment for each study is given in Appendix A.

### 3.4. Impact of Anti-Inflammatory Therapy on Patients with cSDH

#### 3.4.1. Mortality

All included studies reported data on mortality. One thousand and eighty-two patients were randomized into either the anti-inflammatory arm or the placebo arm (542 vs. 540). Forty-two patients in the anti-inflammatory arm became deceased (7.7%), whereas 24 patients became deceased in the placebo arm (4.4%). The period of mortality evaluation ranged from 6 to 12 months after the diagnosis of a cSDH. Figure 3 shows the overall odds ratio (OR) (OR: 1.79; 95% CI: 1.06–3.00) for mortality in the pooled analysis (*p* = 0.03). No significant heterogeneity was present (I^2^ = 0%, *p* = 0.88). 

#### 3.4.2. Neurological Outcome

Four of five included studies reported data on neurological outcome. One thousand and forty-five patients were randomized into either the anti-inflammatory arm or the placebo arm (524 vs. 521). Four hundred and one patients in the anti-inflammatory arm showed a favorable neurological outcome (76.5%) compared to 406 patients in the placebo arm (77.9%). The period of outcome evaluation ranged from 2 to 72 months after the diagnosis. Figure 4 shows the overall odds ratio (OR: 1.12; 95% CI: 0.63–2.00) for favorable outcome in the pooled analysis (*p* = 0.71). No significant heterogeneity was present (I^2^ = 23%, *p* = 0.71). 

#### 3.4.3. Secondary Surgery for Recurrent cSDH

All included studies reported data on the necessity for secondary surgery for either recurrence in primarily surgically treated patients or because of switch of therapy in primarily conservatively treated patients.

One thousand one hundred and seventeen patients were randomized into either the anti-inflammatory arm or the placebo arm (558 vs. 559). Twenty-two patients in the anti-inflammatory arm needed secondary surgery (3.9%), and 24 patients underwent secondary surgery in the placebo arm (10.9%), respectively. Figure 5 shows the overall odds ratio (OR: 0.35; 95% CI: 0.21–0.58) for the need for secondary surgery in the pooled analysis (*p* = 0.0001). No significant heterogeneity was present (I^2^ = 0%, *p* = 0.48). 

### 3.5. Impact of Anti-Inflammatory Therapy on Patients with cSDH Undergoing Primary Surgical Hematoma Evacuation

#### 3.5.1. Mortality

Three of five included studies reported data on mortality in patients undergoing primary surgical hematoma evacuation. Eight hundred and sixty-six patients were randomized into either the anti-inflammatory arm or the placebo arm (434 vs. 432). Thirty-nine patients in the anti-inflammatory arm became deceased (8.9%), whereas 23 patients in the placebo arm (5.3%) became deceased. Figure 6 shows the overall odds ratio (OR: 1.76; 95% CI: 1.03–3.01) for mortality in the pooled analysis (*p* = 0.04). No significant heterogeneity was present (I^2^ = 0%, *p* = 0.82). As far as the primarily surgically treated patients were concerned, all studies showed a superiority of placebo use, compared to use of anti-inflammatory agents, regarding mortality.

#### 3.5.2. Outcome

Three of five included studies reported data on outcome in patients undergoing primary surgical hematoma evacuation. Eight hundred and forty-nine patients were randomized into either the anti-inflammatory arm or the placebo arm (426 vs. 423). Three hundred and six patients in the anti-inflammatory arm showed good outcome (71.8%), whereas 315 patients in the placebo arm (74.5%) had a favorable outcome. Appendix A shows the overall odds ratio (OR: 0.86; 95% CI: 0.63–1.17) for neurological outcome in the pooled analysis (*p* = 0.33). No significant heterogeneity was present (I^2^ = 0%, *p* = 0.40).

#### 3.5.3. Secondary Surgery for Recurrent cSDH

Three of five included studies reported data regarding the need for secondary surgery in patients undergoing primary surgical hematoma evacuation. Nine hundred and one patients were randomized into either the anti-inflammatory arm or the placebo arm (450 vs. 451). Ten patients in the anti-inflammatory arm needed secondary surgery (2.2%), and thirty-five patients in the placebo arm (7.7%) underwent secondary surgery, respectively. Figure 7 shows the overall odds ratio (OR: 0.31; 95% CI: 0.11–0.89) for recurrence in the pooled analysis (*p* = 0.03). No significant heterogeneity was present (I^2^ = 36%, *p* = 0.21).

### 3.6. Impact of Anti-Inflammatory Therapy on Patients with cSDH and Primary Conservative Treatment

#### 3.6.1. Mortality

Two of five included studies reported data on mortality in patients who were treated primarily conservatively. Two hundred and sixteen patients were randomized into either the anti-inflammatory arm or the placebo arm (108 vs. 108). Three patients in the anti-inflammatory arm became deceased (2.7%), whereas one patient became deceased in the placebo arm (0.9%). Figure 8 shows the overall odds ratio (OR: 2.45; 95% CI: 0.35–17.15) for outcome in the pooled analysis (*p* = 0.37). No significant heterogeneity was present (I^2^ = 0%, *p* = 0.39).

#### 3.6.2. Outcome

One of five included studies reported data on outcome in conservatively treated cSDH patients. One hundred and ninety-six patients were randomized into either the anti-inflammatory arm or the placebo arms (98 vs. 98). Ninety-five patients in the anti-inflammatory arm showed a favorable neurological outcome (96.9%), and 91 patients in the placebo arm (92.8%) had a favorable neurological outcome, respectively. Appendix A shows the overall odds ratio (OR: 2.44; 95% CI: 0.61–9.71) for outcome in the pooled analysis (*p* = 0.21). Analysis of heterogeneity was not applicable due to the scarcity of reports.

#### 3.6.3. Secondary Surgery

Two of five included studies reported data regarding the need for secondary surgery in patients who were primarily conservatively treated. Two hundred and sixteen patients were randomized into either the anti-inflammatory arm or the placebo arms (108 vs. 108). Twelve patients in the anti-inflammatory arm needed secondary surgery (11.1%), whereas 26 patients in the placebo arm (24.0%) underwent secondary surgery. Figure 9 shows the overall odds ratio (OR: 0.40; 95% CI: 0.19–0.83) for the need of secondary surgery in the pooled analysis (*p* = 0.01). No significant heterogeneity was present (I^2^ = 0%, *p* = 0.73).

### 3.7. Impact of Corticosteroids on Patients with cSDH

#### 3.7.1. Mortality

Four of five included studies reported data on the mortality of patients treated with corticosteroids. Eight hundred and eighty-six patients were randomized into either the anti-inflammatory arm or the placebo arm (444 vs. 442). Forty-one patients in the anti-inflammatory arm became deceased (9.2%), and 23 patients in the placebo arm (5.2%) became deceased, respectively. Figure 10 shows the overall odds ratio (OR: 1.83; 95% CI: 1.08–3.09) for mortality in the pooled analysis (*p* = 0.03). No significant heterogeneity was present (I^2^ = 0%, *p* = 0.80).

#### 3.7.2. Secondary Surgery

Four of five included studies reported data on secondary surgery in patients treated with corticosteroids. Nine hundred and twenty-one patients were randomized into either the anti-inflammatory arm or the placebo arm (480 vs. 461). Eleven patients in the anti-inflammatory arm needed secondary surgery (2.3%), whereas 38 patients underwent secondary surgery in the placebo arm (8.2%). Figure 11 shows the overall odds ratio (OR: 0.30; 95% CI: 0.14–0.63) for secondary surgery rate in the pooled analysis (*p* = 0.002). No significant heterogeneity was present (I^2^ = 5%, *p* = 0.37).

### 3.8. High-Dose vs. Low Dose Corticosteroids—Mortality

#### 3.8.1. Low Dose Corticosteroids

Two of five included studies reported data on the outcome in patients treated with low-dose corticosteroids. Seven hundred and twenty-six patients were randomized into either the low-dose corticosteroids arm or the placebo arm (363 vs. 363). Thirty-four patients in the low-dose corticosteroid arm became deceased (9.3%), and 19 patients in the placebo arm (5.2%) became deceased, respectively. Figure 12 shows the overall odds ratio (OR: 1.88; 95% CI: 1.05–3.37) for mortality in the pooled analysis (*p* = 0.03). No significant heterogeneity was present (I^2^ = 0%, *p* = 0.77).

#### 3.8.2. High Dose Corticosteroids

Two of five included studies reported data on the outcome in patients treated with high-dose corticosteroids. One hundred and sixty patients were randomized into either the high-dose corticosteroids arm or the placebo arm (81 vs. 79). Seven patients in the high-dose corticosteroid arm became deceased (8.6%), whereas 4 patients in the placebo arm (5.0%) became deceased. Figure 13 shows the overall odds ratio (1.69; 95% CI: 0.51–5.65) for mortality in the pooled analysis (*p* = 0.39). No significant heterogeneity was present (I^2^ = 0%, *p* = 0.36).

### 3.9. Publication Bias

To achieve an acceptable reliability, we took the following three steps to investigate any publication bias: first, the applied literature search strategy was extensive; second, the selected trials in this meta-analysis were strictly in line with the inclusion and exclusion criteria; and finally, third, the publication bias was evaluated by funnel plots (Figure 14) and statistical tests regarding the endpoints (Mortality and recurrence and neurological outcome). The data points were all located inside the inverted funnel, indicating a small publication bias regarding the analysis of the mentioned endpoints.

Subsequently, both Eggar’s and Begg’s tests were performed to rule out a publication bias for all three major outcomes. As far as mortality was concerned, Eggar’s test showed no statistically significant publication bias (*p* = 0.82, intercept = 0.11, 95% CI −1.39–1.61), and Begg’s test showed a Kendall’s tau of 0.40 (*p* = 0.32). Furthermore, Eggar’s test showed no significant publication bias with regard to the endpoint “recurrence” (*p* = 0.59, intercept = −0.63, 95% CI −3.95–2.69) and Begg’s test revealed a Kendall’s tau of 0.20 (*p* = 0.62). As far as neurological outcome was concerned, Eggar’s test revealed no significant publication bias (*p* = 0.99. intercept = −0.005, 95% CI −5.88–5.87), and Begg’s test showed a Kendall’s Tau of 0.00 (*p* = 0.99).

### 3.10. Grading of the Evidence

The certainty of evidence was evaluated as high for all endpoints in the general population, conservatively treated patients, surgically treated patients, and those who underwent corticosteroid treatment. The characteristics and the individual judgments are summarized in Appendix A.

### 3.11. Discussion

In the present meta-analysis, we summarized the evidence from randomized, double-blind, and placebo-controlled trials of anti-inflammatory drug therapy versus placebo for cSDHs. The results of the present meta-analysis were based on the analysis of the following investigational medicinal products (IMP): dexamethasone, prednisone, and atorvastatin. We included five trials that had fulfilled the applied inclusion criteria, with a total number of 1168 patients. 

Our results can be summarized as follows: (1) Anti-inflammatory therapy seems to be associated with increased mortality in surgically treated cSDH patients, whereas it was not found to be associated with increased risk of death in conservatively treated cSDH patients; (2) Anti-inflammatory drug treatment seems to reduce the risk of recurrence, or the probability of a switch to surgical treatment, if it is administered as an adjuvant treatment to surgery or in the case of a conservative regimen; (3) Neurological outcome was not significantly influenced by anti-inflammatory therapy in either surgically treated or conservatively treated cSDH patients; (4) Corticosteroid therapy significantly reduced the recurrence of surgically treated cSDHs, whereas the mortality was paradoxically only associated with the administration of a cumulative low-dose corticosteroid regimen and not the administration of a high cumulative dosage of corticosteroids.

To date, surgical hematoma evacuation via burrhole craniostomy or craniotomy are still the treatment of choice for cSDH patients. However, surgical hematoma removal is still associated with a high rate (10–25%) of secondary surgery for recurrent cSDH [43,44,45]. Furthermore, conservatively treated cSDH patients without neurological deficits necessitating surgery are also at risk of cSDH progression. Therefore, anti-inflammatory therapy might be an option to reduce the risk of cSDH progression necessitating a switch to surgical therapy. 

Inflammatory response is a known mechanism facilitating the building of the external membrane in cSDH and is suggested to be a source of cSDH growth [8]. Hence, the attenuation of inflammatory reactions and neoangiogenesis might be novel avenues to enhance cSDH resorption. Inhibition of the 3-hydroxy-3-methylglutarzyl-coenzyme A reductase by atorvastatin was found to reduce the cSDH volume [46]. Statins can induce angiogenesis and increase the amount of peripheral blood endothelial progenitor cells by activating the Akt pathway, Notchl pathway, and endothelial nitric oxide synthase [47,48,49,50,51]. The number of endothelial progenitor cells is significantly lower in cSDH patients compared to healthy people, and the level of endothelial progenitor cells is also significantly lower in cSDH patients with recurrence compared to those without a recurrent cSDH [50]. Endothelial progenitor cells are essential for the repair of damaged vascular endothelium [51]. Atorvastatin has an anti-inflammatory function in its the mobilization of endothelial progenitor cells to repair vascular damage [52]. Furthermore, atorvastatin inhibits inflammatory angiogenesis through down regulation of VEGF, TNF-alpha and TGF-beta1 [52]. Li et al. [53] microscopically examined SDHs in rats and found that atorvastatin treatment significantly reduced the neutrophil counts in the neomembranes. A recent meta-analysis of 6 (prospective and retrospective) cSDH studies found that atorvastatin could improve both the rate of recurrence and neurological functioning [49].

As far as the progression of cSDH volume is concerned, the results of this meta-analysis were in line with our results. However, we included only one prospective, randomized, double-blind, and placebo-controlled trial by Jiang et al. [24], investigating atorvastatin in conservatively treated cSDH patients. The rate of cSDH progression with the need of a switch to surgery was significantly lower in those patients allocated to the atorvastatin arm, compared to the placebo group. However, we could not confirm the improvement of neurological functioning by atorvastatin treatment. Nevertheless, it has to be remembered that our result with regard to atorvastatin was based on only one study investigating conservatively treated patients, who are generally predominantly asymptomatic patients. Several studies suggested that atorvastatin might be a potential alternative to surgical evacuation in patients with no, or mild, symptoms [54,55,56]. Future trials investigating atorvastatin should focus on its role as an adjuvant therapy for those who underwent surgery, and more trials are needed to elucidate the impact of atorvastatin on neurological functioning in conservatively treated patients.

Injury to the dural border cells induces a release of inflammatory cytokines, which enhances the recruitment of macrophages, formation of granulation tissue, and vascular repair mechanisms (see Appendix A). Afterwards, novel thin capillaries are formed, and they are prone to continuous microhemorrhages, due to their high permeation [8,33]. Corticosteroids, as anti-inflammatory and anti-angiogenic therapy, might reduce the rate of recurrence and progression of cSDH [57]. Synthetic corticosteroids, such as dexamethasone, can inhibit the recruitment of macrophages, phagocytosis, and secretion of inflammatory cytokines. Furthermore, it has been demonstrated that dexamethasone significantly reduced the size and weight of blood clots in the fluid collection of cSDH, due to its anti-inflammatory properties attenuating the formation of the external membrane [57]. Our findings suggested that corticosteroids could reduce both the rate of recurrence in surgically treated patients as well as the rate of switch to surgery in conservatively treated cSDH patients. This finding, regarding the reduction of recurrence, was in line with two other meta-analyses [58,59]. Anti-inflammatory drug therapy resulted in an increased mortality in surgically treated patients. However, this effect of anti-inflammatory drug therapy could not be found in the analysis of only conservatively treated patients. Furthermore, the effect of an increased mortality rate in cSDH patients treated by corticosteroids was paradoxically found to be based on the effect of a low cumulative dosage corticosteroid regimen. However, this comparison, regarding the cumulative dosage, should be interpreted with caution because in low-dose studies by Hutchinson et al. [22] and Mebberson et al. [23] the patients were predominantly treated by surgery with the exception of 38 patients in the trial by Hutchinson et al. [22]. Conversely, in our comparison of high-dosage corticosteroid studies, compared to their placebo arm, significantly more patients underwent a conservative regimen. However, the high-dose application of corticosteroids, as well as atorvastatin therapy, resulted in no significant increased mortality. The use of dexamethasone or corticosteroids in cSDH patients is highly debated because of their potentially severe side effects which could outrank the advantages regarding the attenuation of cSDH growth. A network meta-analysis found that dexamethasone increased the risk of all-cause mortality of cSDH with a relative risk of 1.96 [60]. However, our results indicated the need for a further trial investigating the effect of high-dose corticosteroid therapy in conservatively treated cSDH patients. Interestingly, a randomized trial comparing the combination of low-dose dexamethasone with atorvastatin found a more significant reduction of the hematoma volume in conservatively treated patients by the combination schedule (low-dose dexamethasone with atorvastatin) compared to atorvastatin alone [61]. A further explorative study using mass spectrometry of the hematoma fluid and macrophages culture revealed that dexamethasone in combination with atorvastatin increased the levels of atorvastatin in the hematoma and induced the transition from the M1 (pro-inflammatory) macrophage phenotype towards the M2 (anti-inflammatory) macrophage phenotype [62]. Hence, the drugs could have synergistic functioning and dexamethasone combined with atorvastatin could facilitate anti-inflammatory functions, which induce vascular repair in order to reduce the risk of continuous micro-bleeding into the hematoma collection. However, this potential combination therapy for conservatively treated patients necessitates further validation by means of a phase III with a double-blind and placebo-controlled design.

According to the results, mild/asymptomatic patients with cSDH treated conservatively seemed to be a subgroup potentially profiting from optimally administered drugs or a combination of drugs. Particularly elderly patients with clinically relevant cerebral atrophy might tolerate a higher volume of the hematoma and remain asymptomatic. In such patients, an omittance of surgical hematoma evacuation might spare potentially fatal complications of the surgery [63,64] and enhance the neurological outcome [65], if clinically tenable. However, due to the non-standardized treatment algorithm, patients undergoing conservative therapy might not reach the neurosurgical specialist, due to outpatient setting, and remain underrepresented, even in prospective neurosurgical trials. Conservatively treated patients might represent a suitable cohort for further investigation.

#### Limitations

This meta-analysis demonstrated a potential field for anti-inflammatory drug therapy in conservatively treated cSDH patients with regard to the reduction of the switch to surgery while, simultaneously, not influencing mortality. The present meta-analysis applied high-selective inclusion criteria by investigating prospective, randomized, double-blind, and placebo-controlled trials only. Hence, the most important advantage of the present meta-analysis is the summary of all eligible studies fulfilling this low-risk of bias criteria to draw conclusions regarding the effectiveness of anti-inflammatory therapies in cSDH patients. However, the conclusions were limited by the fact that individual trials investigated anti-inflammatory drug therapy in both surgical and conservatively treated cSDH patients. Furthermore, the results of the present meta-analysis were based on only five prospective randomized, placebo-controlled and double-blinded trials. Hence, there were still not enough level 1 randomized, placebo-controlled, and double-blinded studies on statins in either surgically or conservatively treated cSDH patients. Despite the high number of patients cumulatively included in the analysis of each outcome, the main limitation of this study was the scarcity of the RCTs performed on patients suffering cSDH. Furthermore, the eligible studies differed in the observational period and outcome measures reported, which was the main limitation in the “neurological outcome” analysis. Furthermore, we did not investigate adverse drug reactions of the anti-inflammatory drug therapies, which might also have clinical implications for further trials.

## 4. Conclusions

Anti-inflammatory drug therapy significantly reduced the recurrence and growth of cSDH in surgically and conservatively treated cSDH patients. Corticosteroid therapy significantly increased mortality in surgically treated patients, whereas anti-inflammatory therapy (atorvastatin or corticosteroid) therapy did not increase mortality in conservatively treated cSDH patients. Neurological outcome was not influenced by anti-inflammatory therapy in surgically or conservatively treated cSDH patients. Further multi-centric, randomized, double-blind and placebo-controlled trials are needed to establish sufficient evidence regarding anti-inflammatory mono-therapies and combination therapies with the main focus being on the promising subgroups.

## Figures and Tables

**Figure 1 ijms-23-16198-f001:**
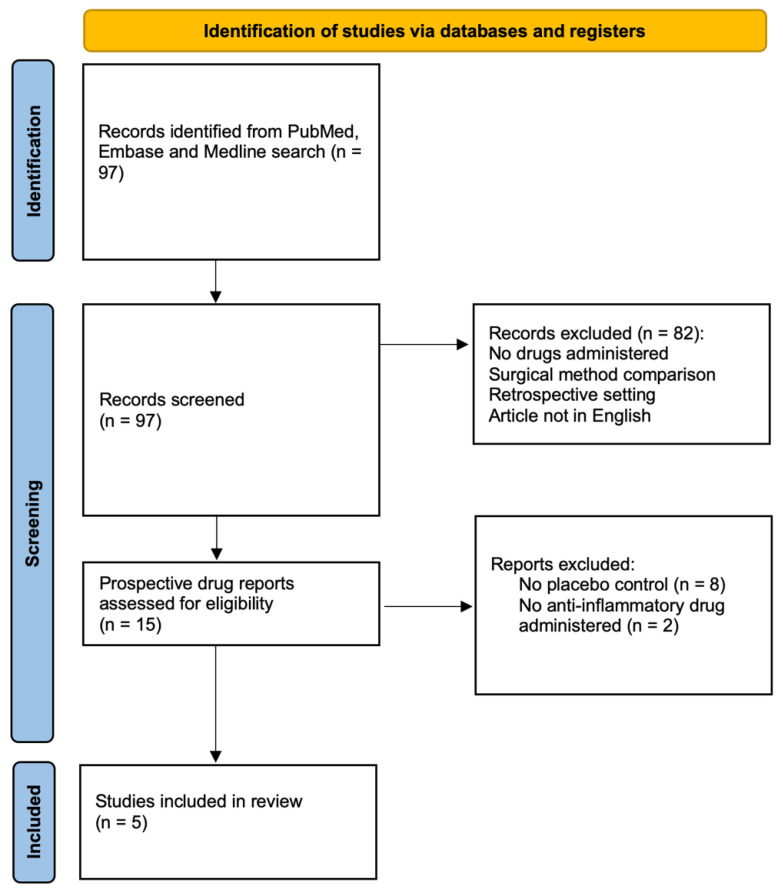
PRISMA flow chart illustrating the study selection.

**Figure 2 ijms-23-16198-f002:**
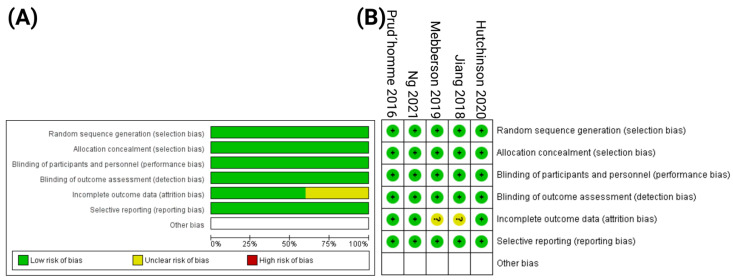
(**A**) Risk of bias assessment for each kind of bias. (**B**) Summary of risk of bias of the included randomized controlled trials (reviewers’ judgments about each risk of bias characteristic of the included trials: “+” constitutes low risk; “?” constitutes unclear risk).

**Figure 3 ijms-23-16198-f003:**
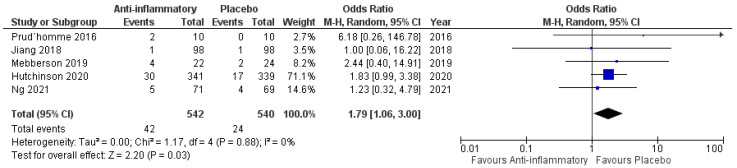
Forest Plots displaying OR and 95% CI estimates for mortality in studies [28,29,30,31,32] evaluating anti-inflammatory therapies compared to placebo in cSDH patients treated surgically or conservatively. Squares represent the odds ratio; the bigger the square, the greater the weight given because of the narrower 95% CI. Diamond represents the odds ratio of the overall data.

**Figure 4 ijms-23-16198-f004:**
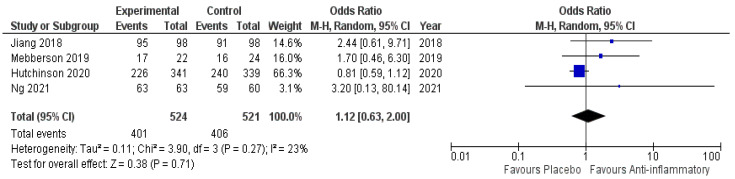
Forest Plots displaying OR and 95% CI estimates for favorable neurological outcome in studies [28,29,30,31] evaluating anti-inflammatory therapies compared to placebo in cSDH patients treated surgically or conservatively. Squares represent the odds ratio; the bigger the square, the greater the weight given because of the narrower 95% CI. Diamond represents the odds ratio of the overall data.

**Figure 5 ijms-23-16198-f005:**
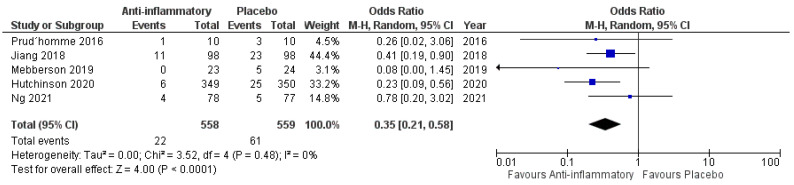
Forest Plots displaying OR and 95% CI estimates for the need of secondary surgery in studies [28,29,30,31,32] evaluating anti-inflammatory therapies compared to placebo in cSDH patients treated surgically or conservatively. Squares represent the odds ratio; the bigger the square, the greater the weight given because of the narrower 95% CI. Diamond represents the odds ratio of the overall data.

**Figure 6 ijms-23-16198-f006:**
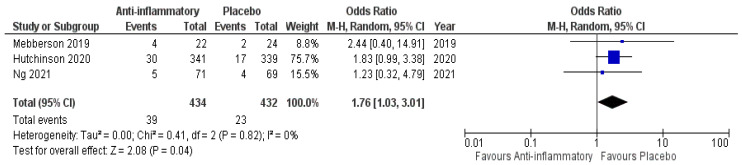
Forest Plots displaying OR and 95% CI estimates for mortality in studies [28,29,30] evaluating anti-inflammatory therapies compared to placebo in patients who underwent surgical treatment of cSDH. Squares represent the odds ratio; the bigger the square, the greater the weight given because of the narrower 95% CI. Diamond represents the odds ratio of the overall data.

**Figure 7 ijms-23-16198-f007:**
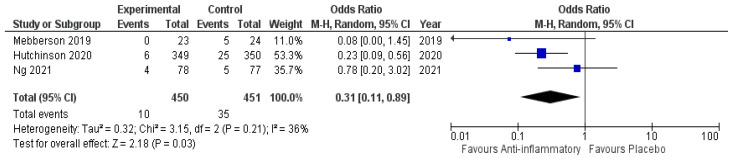
Forest Plots displaying OR and 95% CI estimates for the need of secondary surgery in studies [28,29,30] evaluating anti-inflammatory therapies compared to placebo in patients who underwent surgical treatment of cSDH. Squares represent the odds ratio; the bigger the square, the greater the weight given because of the narrower 95% CI. Diamond represents the odds ratio of the overall data.

**Figure 8 ijms-23-16198-f008:**
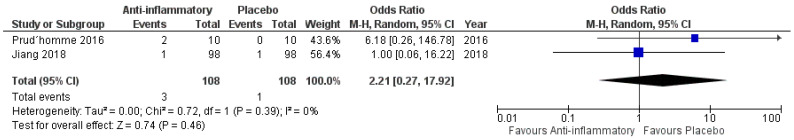
Forest Plots displaying OR and 95% CI estimates for mortality in studies [31,32] evaluating anti-inflammatory therapies compared to placebo in patients with cSDH treated conservatively. Squares represent the odds ratio; the bigger the square, the greater the weight given because of the narrower 95% CI. Diamond represents the odds ratio of the overall data.

**Figure 9 ijms-23-16198-f009:**
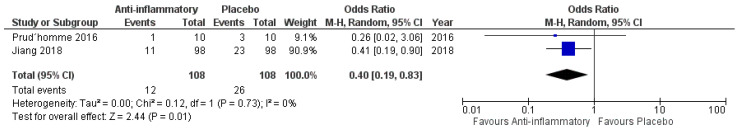
Forest Plots displaying OR and 95% CI estimates for the need of secondary surgery in studies [31,32] evaluating anti-inflammatory therapies compared to placebo in patients who underwent conservative treatment of cSDH. Squares represent the odds ratio; the bigger the square, the greater the weight given because of the narrower 95% CI. Diamond represents the odds ratio of the overall data.

**Figure 10 ijms-23-16198-f010:**
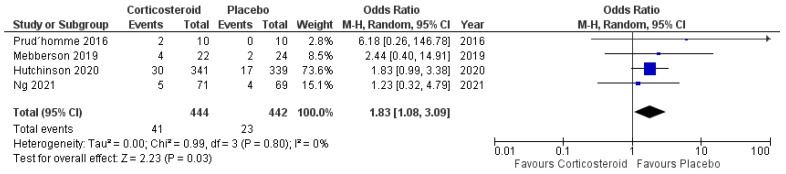
Forest Plots displaying OR and 95% CI estimates for mortality in studies [28,29,30,32] evaluating anti-inflammatory therapies compared to placebo in patients with cSDH. Squares represent the odds ratio; the bigger the square, the greater the weight given because of the narrower 95% CI. Diamond represents the odds ratio of the overall data.

**Figure 11 ijms-23-16198-f011:**
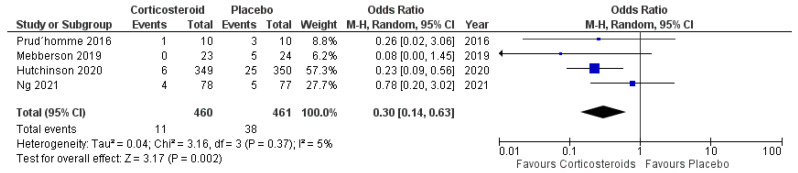
Forest Plots displaying OR and 95% CI estimates for secondary surgery for progression or recurrence in studies [28,29,30,32] evaluating anti-inflammatory therapies compared to placebo in patients with cSDH. Squares represent the odds ratio; the bigger the square, the greater the weight given because of the narrower 95% CI. Diamond represents the odds ratio of the overall data.

**Figure 12 ijms-23-16198-f012:**
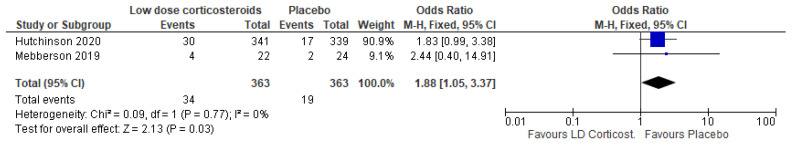
Forest Plots displaying OR and 95% CI estimates for mortality in studies [29,30] evaluating the use of low-dose corticosteroids compared to placebo in patients with cSDH. Squares represent the odds ratio; the bigger the square, the greater the weight given because of the narrower 95% CI. Diamond represents the odds ratio of the overall data.

**Figure 13 ijms-23-16198-f013:**
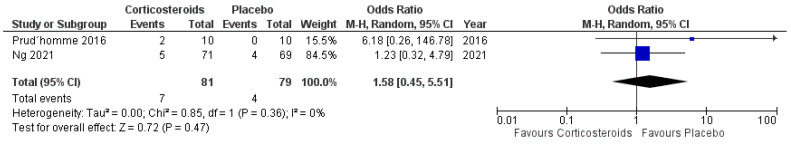
Forest Plots displaying OR and 95% CI estimates for mortality in studies [28,32] evaluating the use of high-dose corticosteroids compared to placebo in patients with cSDH. Squares represent the odds ratio; the bigger the square, the greater the weight given because of the narrower 95% CI. Diamond represents the odds ratio of the overall data.

**Figure 14 ijms-23-16198-f014:**
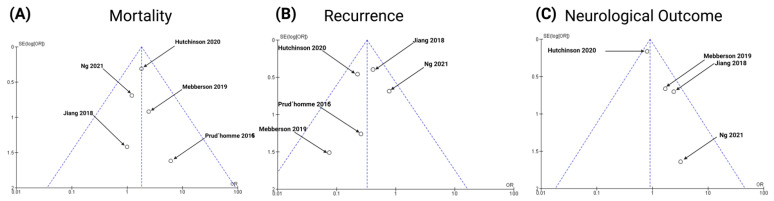
Funnel plots for the following endpoints of the present meta-analysis [28,29,30,31,32]: Mortality (**A**), Recurrence (**B**), Neurological outcome (**C**). The midline of the studies indicates no publication bias of studies comparing anti-inflammatory therapies with conventional therapy.

**Table 1 ijms-23-16198-t001:** Major characteristics of anti-inflammatory studies included in the present meta-analysis.

Name	Year	Treatments and Dosage	Sample Size (n)	Intervention (n)	Control (n)	Reported Outcome	Country & Number of Centers
Ng et al. [28]	2021	Prednisone 1 mg/kg/d	155	78	77	Markwalder grading score (MGS) at 12 months	France (Multicentric)
Hutchinson et al. [29]	2020	Dexamethasone 8 mg 2× daily	748	375	373	Modified Rankin scale (mRS) at 6 months	United Kingdom (Multicentric)
Mebberson et al. [30]	2019	Dexamethasone 128 mg/2 weeks	47	24	22	Modified Rankin scale (mRS) at 6 months	Australia (Monocentric)
Jiang et al. [31]	2018	Atorvastatin 20 mg/day for 8 weeks	196	98	98	Markwalder grading score (MGS) at 12 months	China (Multicentric)
Prud´homme et al. [32]	2016	Dexamethasone 12 mg/day for 3 weeks	20	10	10	Mortality at 6 months	Canada (Monocentric)

## Data Availability

All data were inncluded in this manuscript.

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
