# Peer review of "Anti-Inflammatory Drug Therapy in Chronic Subdural Hematoma: A Systematic Review and Meta-Analysis of Prospective Randomized, Double-Blind and Placebo-Controlled Trials"

_ijms, 2022, doi:10.3390/ijms232416198_

Round 1

Reviewer 1 Report

1) First and foremost, the topic may be interesting, but its lack of exploration of neuroinflammatory mechanisms does not fit the requirements of the International Journal of Molecular Sciences and this special issue.

2) The lack of a literature quality assessment is a methodological mistake and may lead to unreliable results due to the limited literature included. I would suggest the Cochrane ROB tool, which is known to assess the quality of randomized controlled trials.

3)Prud'homme 2016 (2016, not 2015) included only 20 patients, and the design was not rigorous enough to meet the criteria for a randomized controlled study. Whether to include this experiment needs to be confirmed by evidence after quality assessment.

4)Atorvastatin is both a localized inflammation inhibitor and an angiogenesis maturation agent. I wonder whether it is appropriate to classify it as an anti-inflammatory agent and compare it with glucocorticoids.

5)Mebberson 2019 is just an interim analysis. Maybe the final results of the trial would be more valid.

6) As shown in Figure 1, both the first excluded step (N=83) and the second excluded step (N=9) included “No anti-inflammatory drug administered”. Please explain the difference between these two exclusions and why they are divided into two steps.

Author Response

Dear Reviewer

Thank you for reading our manuscript and critically reviewing it, which will help us improve it to a better scientific level and make it more understandable to the readership.

In the following we would like to respond to your remarks:

We agree with the reviewer that the manuscript necessitates a detailed exploration of the role neuroinflammation and statins in cSDH. Therefore, we made the following revisions regarding this issue:

  1. Introduction: HMG-CoA reductase inhibitors were found to enhance the resoption of hematomas. Furthermore, statins can reduce the brain edema by increasing the amount of Treg cells in the central nervous system. Treg cells are known anti-inflammatory regulators [1].
  2. Discussion: Atorvastatin can inhibit the pro-inflammatory angiogenesis by downregulating the expression of VEGF, TNF-alpha, and TGF-beta [2]. Li et al. [3] performed microscopical examinations of cSDH neomembranes in rats and revealed that atorvastatin significantly reduces the neutrophil counts in the neombranes [3]. Additionally, they found that atorvastatine treatment significantly decreased the expression and secretion of TNF-alpha, IL-6, and VEGF.
  3. Supplementary figure S3: We also implemented the extended review on the anti-inflammatory role of atorvastatin in the supplementary figure S3 in order to further facilitate the understanding.

The reviewer is absolutely right that a literature quality assessment is essential to be included in the manuscript. Therefore, we created the section “2.3 Data extraction and quality evaluation” in the methods. Study names, first authors, year of publication, country, number of centers (mono-, bi- or multicentric), basic trial design (randomization, allocation, lack of data, lack of reported outcome, intention-to-treat analysis), and other relevant data were extracted as baseline data. The Cochrane Bias Risk Tool was used to investigate the risk of bias (ROB) in the included trials using the software Review Manager Web (RevMan Web Version 5.4.1 from The Cochrane Collaboration, available at revman.cochrane.org). The following six characteristics regarding risk of bias assessment were included in the analysis: selection bias, performance bias, detection bias, attrition bias, reporting bias, and other sources of bias. Afterwards, a risk of bias summary chart and plot were created. Furthermore, the section “2.5 Grading of the evidence” was also added to the methods. The overall certainty of all trials was investigated using Grading of Recommendations, Assessment, Development, and Evaluation (GRADE) [4, 5]. Criteria to judge included trial limitations (according to Newcastle Ottawa Scale [6]), inconsistency (significant study heterogeneity, I2 >50%), indirectness (characteristics limiting the generalizability of the findings), imprecision (the 95% CI for effect estimates crosses for an essential difference of 5% from the line of unity), and publication bias (small trial effects causing significant evidence). The results of the risk of bias quality assessment were added to the newly created section “3.3 Risk of Bias Quality Assessment” in the results. The results are summarized in a newly created figure 2. Our protocol summarizing the literature quality assessment workflow is provided in the newly created supplementary table S2. Furthermore, publication bias was assessed using Eggar´s and Begg´s test (section “3.9 Publication bias”). Both statistical tests analyzing publication bias revealed no significant asymmetry of our funnel plots. Results of the grading of the certainty of evidence are provided in the section “3.9 Grading of the evidence”. The certainty of evidence was evaluated as high for all endpoints in the general population, conservatively treated patients, surgically treated patients, and those who underwent corticosteroid treatments. The findings and individual author judgments are summarized in the supplementary tables S3-S6.

The reviewer is absolutely right that the online version of Prud´homme et al. [7] was released in 2016. Hence, we replaced the submission date (2015) by the publication date (2016) in all figures (Figures 3, 5, 8, 9, 10, 11, 13, and 14) displaying forest plots of this trial.

Furthermore, we agree with the reviewer that Mebberson et al. [8] revealed only the results their interim analysis so far. However, the final results are not yet published and the present manuscript of Mebberson et al. [8] fulfilled the criteria of a prospective randomized, placebo-controlled, and double-blinded trial.

We thank the reviewer regarding the remark about figure 1. We apologize for this mistake. We screened our search flow again and identified this mistake, too. In the first exclusion of records it should have been stated “no drugs administered”. Afterwards, in the second exclusion step we screened all prospective drug reports and removed those who did not investigate anti-inflammatory drugs or used no placebo controls. Schaumann et al. [9] analyzed the use of celecoxib in a randomized cSDH trial without a placebo control and was also placed to the second exclusion step. Hence, the number of the first and second round of exclusion changed by one record. All excluded prospective trials on drugs (n =10: 8 trials without a placebo control and 2 trials without an anti-inflammatory drug) in cSDH patients are summarized in the supplementary table S1.

References

  1. Quan, W.; Zhang, Z.; Li, P.; Tian, Q.; Huang, J.; Qian, Y.; et al. Role of regulatroy T cells in atorvastatin induced absorption of chronic subdural hematoma in rats. Aging Dis. 2019, 10, 992-1002
  2. Araujo, F.; Rocha, M.A.; Mendes, J.B.; Andrade, S.P. Atorvastatin Inhibits Inflammatory Angiogenesis in Mice through Down Regulation of VEGF, TNF-Alpha and TGF-Beta1. Biomed Pharmacother. 2010, 64(1), 29-34
  3. Li, T.; Wang, D.; Tian, Y.; Yu, H.; Wang, Y.; Quan, W.; Cui, W.; Zhou, L.; Chen, J.; Jiang, R.; Zhang, J. Effects of atorvastatin on the inflammation regulation and elimination of subdural hematoma in rats. J Neruol Sci. 2014, 341(1-2), 88-96
  4. Balshem, H.; Helfand, M.; Schünemann, H.J.; Oxman, A.D.; Kunz, R.; Brozek Jet, et al. GRADE guidelines: 3. Rating the quality of evidence. J Clin Epidemiol. 2011, 64(4), 401-6
  5. Schünemann, H.; Brozek, J.; Guyatt, G.; Oxman, A.; editors. GRADE handbook for grading quality of evidence and strength of recommendations (updated October 2013) [internet]: The GRADE Working Group. 2013. Available online: https://gdt.gradepro.org/app/handbook/handbook.html
  6. Wells, G.A.; Shea, B.; O´Connell, D.; Petersen, J.; Welch Vet, et al. The Newcastle-Ottawa Scale (NOS) for assessing the quality of nonrandomized studies in meta-analyses. [internet]. Ottawa: Ottawa Hospital Research Institute. 2000. Available online: https://www.ohri.ca/programs/clinical_epidemiology/oxford.asp
  7. Prud'homme, M., Mathieu, F., Marcotte, N., Cottin, S. A Pilot Placebo Controlled Randomized Trial of Dexamethasone for Chronic Subdural Hematoma. Can J Neurol Sci. 2016, 43(2), 284-90
  8. Mebberson, K., Colditz, M., Marshman, L.A.G., Thomas, P.A.W., Mitchell, P.S., Robertson, K. Prospective randomized placebo-controlled double-blind clinical study of adjuvant dexamethasone with surgery for chronic subdural haematoma with post-operative subdural drainage: Interim analysis. J Clin Neurosci. 2020, 71, 153-157
  9. Schaumann, A.; Klene, W.; Rosenstengel, C.; Ringel, F.; Tüttenberg, J.; Vajkoczy, P. COXIBRAIN: results of the prospective, randomized, phase II/III study for the selective COX-2 inhibition in chronic subdural haematoma patients. Acta Neurochir (Wien). 2016, 158(11), 2039-2044

Reviewer 2 Report

The authors investigated the impact of anti-inflammatory drug therapy on mortality and outcome in chronic subdural hematoma(cSDH); Their results illustrated significantly lower rates of recurrent cSDH (OR: 0.35; 95 % CI: 0.21-0.58, p=0.0001) in patients undergoing anti-inflammatory therapy. Anti-inflammatory therapy was associated with lower rates of “switch-to-surgery” cases (OR: 0.30; 95 % CI: 0.14-0.63, p=0.002). This meta-analysis demonstrates a potential field for anti-inflammatory drug therapy in cSDH patients with regard to the reduction of the switch to surgery and simultaneously not influencing the mortality.

Comments:

Abstract 

1. PRISMA checklist asks the author to clarify the systematic review registration number in Structured summary. Do you have a registration number, such as Cochrane or PROSPERO? Please provide your registration number; I think meta-analysis with a registration number can reduce reporting bias. 

2. Do the authors strictly follow the PRISMA checklist? If so, the Checklist item and reported on which page should list as supplementary materials (you can refer to PRISMA 2009 Checklist).

3. The effect of dexamethasone on outcomes in patients with chronic subdural hematoma resulted in fewer favorable outcomes and more adverse events; I did a cursory search, and you omitted some RCTs regarding dexamethasone in patients with CSDH. You can refer to an article (Yu W, Chen W, Jiang Y, et al. PMID: 35401183). Moreover, there are many anti-inflammatory drugs, such as Selective COX-2 Inhibition (Schaumann et al., 2016, PMID: 27605230). Dexamethasone has limited efficacy in chronic subdural hematoma, and there have been many meta-analyses of it; Atorvastatin may be a safe and efficacious nonsurgical alternative for treating patients with CSDH; please explain the reason for your classification.

Introduction  

1. Due to the global population aging and spreading of slice imaging, the overall incidence increased significantly from 8.2 to 17.6 per 100,000/year between 1990 and 2015……please use the incidence of cSDH in the past three years.

Materials and Methods  

1. the Cochrane Handbook for Systematic Reviews of Interventions Version 5.1.0…. Please correct.

2. Why only search Pubmed, Embase and Medline? (The databases retrieved in the abstract are PubMed, Cochrane Library and Embase), please check.

3. Please supply the start and end time of the search literature in the Inclusion and exclusion criteria.

4. Are there other search terms besides "chronic subdural hematoma"? It is recommended that meta-analysis be determined according to the PICOS principle, and it is best to provide search strategy, search results of each database in the supplementary materials. 

5. Using a funnel plot to judge publication bias is too subjective; Egger's Test and Begg's Test are recommended.

Results

1. What are the Neurological outcomes mainly? Please describe in detail.

2. There are too many figures; please put part of the Figure in supplementary materials or a combination in main context.

3. Please list the strengths and weaknesses of your research in proper context.

4. I recommend that they check their English writing.

Author Response

Dear Reviewer

Thank you for reading our manuscript and critically reviewing it, which will help us improve it to a better scientific level and make it more understandable to the readership.

In the following we would like to respond to your remarks:

We agree with the reviewer that the PRISMA checklist is an important tool regarding systematic reviews. Hence, we have now added the checklist to the manuscript. Due to the high number of figures, we inserted the checklist to the supplementary material. Supplementary figure S1 shows the PRISMA checklist in adherence to Moher et al. [1].

Furthermore, we started the submission to PROSPERO and noticed that it is not possible to retrospectively register this meta-analysis in PROSPERO (see attached file). Therefore, we cannot register the meta-analysis at this stage and we do not want to register an already submitted manuscript because this would be not be in line with good clinical practice. We apologize for this mistake and we strive to register further analyses prior to the submission.

The reviewer is absolutely right that dexamethasone reduces the recurrence but dexamethasone has negative effects regarding neurological outcome and the frequency of adverse events. The articles provided in the meta-analysis by Yu et al. [2] were also screened in our search workflow and did not fulfill our inclusion criteria. Sun et al. [3] evaluated dexamethasone in a cSDH cohort of patients with four different treatment methods (conservative treatment + dexamethasone, surgery + dexamethasone, surgery without dexamethasone, and conservative without dexamethasone). However, they did not use a placebo control and there is no randomization method described. Furthermore, Wang et al. [4] compared the combination of dexamethasone and atorvastatin with atorvastatin only. This trial has no placebo control and is rather a proof of concept trial regarding the combined use of dexamethasone and atorvastatin. Schaumann et al. [5] investigated the selective COX-2 inhibitor celecoxib in subdural hematomas. This interesting study was not included because this trial had no placebo control. The supplementary table S1 summarizes the excluded prospective randomized clinical trials on anti-inflammatory drugs in cSDH.

The incidence of cSDH was updated by a study published by Stubbs et al. [6] in 2021. This study from Cambridge in the UK reports an incidence ranging from 8.2/100,000/year to 48/100,000/year in the years 2015-2018.

The version of the Cochrane Handbook for Systematic Reviews of Interventions was updated to Version 6.3 [7]. The databases Pubmed, Medline, and Embase were screened using our inclusion and exclusion criteria until the 30th of June 2022.

The literature search included all results until the 30th of June 2022. The inclusion criteria were formulated according to the PICOS (population, intervention, comparator, outcomes and study design) framework [8].These criteria were as follow: patients had undergone treatment for chronic subdural hematoma; relevant anti-inflammatory drug therapies were performed; results were compared to a placebo control and all results of the prespecified endpoints are reported; and the trials were defined as prospective randomized, placebo-controlled, and double-blinded studies. The following types of records were excluded: reviews, study protocols, letters, conference abstracts, unpublished papers, animal experiments, and trials with insufficient data (e.g., no placebo control or not double-blinded).

The reviewer is absolutely right that a literature quality assessment is essential to be included in the manuscript. Therefore, we created the section “2.3 Data extraction and quality evaluation” in the methods. Study names, first authors, year of publication, country, number of centers (mono-, bi- or multicentric), basic trial design (randomization, allocation, lack of data, lack of reported outcome, intention-to-treat analysis), and other relevant data were extracted as baseline data. The Cochrane Bias Risk Tool was used to investigate the risk of bias (ROB) in the included trials using the software Review Manager Web (RevMan Web Version 5.4.1 from The Cochrane Collaboration, available at revman.cochrane.org). The following six characteristics regarding risk of bias assessment were included in the analysis: selection bias, performance bias, detection bias, attrition bias, reporting bias, and other sources of bias. Afterwards, a risk of bias summary chart and plot were created. Furthermore, the section “2.5 Grading of the evidence” was also added to the methods. The overall certainty of all trials was investigated using Grading of Recommendations, Assessment, Development, and Evaluation (GRADE) [9, 10]. Criteria to judge included trial limitations (according to Newcastle Ottawa Scale [11]), inconsistency (significant study heterogeneity, I2 >50%), indirectness (characteristics limiting the generalizability of the findings), imprecision (the 95% CI for effect estimates crosses for an essential difference of 5% from the line of unity), and publication bias (small trial effects causing significant evidence). The results of the risk of bias quality assessment were added to the newly created section “3.3 Risk of Bias Quality Assessment” in the results. The results are summarized in a newly created figure 2. Our protocol summarizing the literature quality assessment workflow is provided in the newly created supplementary table S2. Furthermore, publication bias was assessed using Eggar´s and Begg´s test (section “3.9 Publication bias”). Egger´s and Begg´s test were performed using MedCalc (Version 20.123 for Windows). Both statistical tests analyzing publication bias revealed no significant asymmetry of our funnel plots. Results of the grading of the certainty of evidence are provided in the section “3.9 Grading of the evidence”. The certainty of evidence was evaluated as high for all endpoints in the general population, conservatively treated patients, surgically treated patients, and those who underwent corticosteroid treatments. The findings and individual author judgments are summarized in the supplementary tables S3-S6.

Favorable neurological outcome was assessed using either modified ranking scale or the Markwalder grading score in the included trials. Modified Ranking scale was used to measure the neurological outcome, which was subsequently evaluated in accordance with the definition of poor outcome as a score of 3-6 and good outcome as a score of 0-2 [12]. Markwalder grading score was dichotomized into good (0-1) and poor (2-4) outcome. Four of the 5 included trials reported on neurological outcome in Markwalder grading score (MGS) or modified Rankin scale [13-16]. Unfortunately, a more detailed statistical analysis of anti-inflammatory drugs regarding outcome of neurological deficits (e.g., degree of palsies, cranial nerve dysfunctions) is not possible in a sufficient scientific way.

We decreased the number of figures by adding two forest plots (Supplementary Figure S2 & S3) to the section “Supplementary material”.

We absolutely agree with the reviewer that it is of paramount importance to critically discuss the strengths and weaknesses of the present meta-analysis. Hence, we created the section “4.1 Limitations”. This meta-analysis demonstrates a potential field for anti-inflammatory drug therapy in conservatively treated cSDH patients with regard to the reduction of the switch to surgery and simultaneously not influencing the mortality. The present meta-analysis applied high-selective inclusion criteria by investigating prospective, randomized, double-blind, and placebo-controlled trials only. Hence, the most important advantage of the present meta-analysis is the summary of all eligible studies fulfilling this low-risk of bias criteria to draw conclusions regarding the effectiveness of anti-inflammatory therapies in cSDH patients. However, the conclusions are limited by the fact that individual trials investigated anti-inflammatory drug therapy in both surgical and conservatively treated cSDH patients.  Hence, there are still not enough level 1 randomized, placebo-controlled, and double-blinded studies on statins in either surgically or conservatively treated cSDH patients. Despite high number of patients cumulatively included in the analysis of each outcome, the main limitation of this study is the scarcity of the RCTs performed on patients suffering on cSDH. Furthermore, the eligible studies differ in the observational period and outcome measures reported, which is the main limitation in the “neurological outcome” analysis. Furthermore, we did not investigate adverse drug reactions of the anti-inflammatory drug therapies, which might also have clinical implications for further trials.

Finally, the manuscript underwent substantial language editing by an native medical professional.

References

  1. Moher, D.; Liberati, A.; Tetzlaff, J.; Altman, D.G. Preferred reporting items for systematic reviews and meta-analyses: the PRISMA statement. PLoS Med. 2009, 6, e1000097
  2. Yu, W.; Chen, Q.; Jiang, Y.; Ma, M.; Zhang, W.; Zhang, X.; Cheng, Y. Effectiveness Comparisons of Drug Therapy on Chronic Subdural Hematoma Recurrence: A Bayesian Network Meta-Analysis and Systematic Review. Front Pharmacol. 2022, 13, 845386
  3. Sun, T.F., Boet, R., Poon, W.S. Non-surgical primary treatment of chronic subdural haematoma: Preliminary results of using dexamethasone. Br J Neurosurg. 2005, 19(4), 327-33
  4. Wang, D.; Gao, C.; Xu, X.; Chen, T.; Tian, Y.; Wie, H.; Zhang, S.; Quan, W.; Wang, Y.; Yue, S.; Wang, Z.; Lei, P.; Anderson, C.; Dong, J.; Zhang, J.; Jiang, R. Treatment of chronic subdural hematoma with atorvastatin combined with low-dose dexamethasone: phase II randomized proof-of-concept clinical trial. J Neurosurg. 2020, 31, 1-9
  5. Schaumann, A.; Klene, W.; Rosenstengel, C.; Ringel, F.; Tüttenberg, J.; Vajkoczy, P. COXIBRAIN: results of the prospective, randomized, phase II/III study for the selective COX-2 inhibition in chronic subdural haematoma patients. Acta Neurochir (Wien). 2016, 158(11), 2039-2044
  6. Stubbs, D.J.; Vivian, M.E.; Davies, B.M.; Ercole, A.; Burnstein, R.; Joannides, A.J. Incidence of chronic subdural haematoma: a single-centre exploration of the effects of an ageing population with a review of the literature. Acta Neurochir (Wien). 2021. 163(9), 2629-2637
  7. Higgins, J.P.T.; Thomas, J.; Chandler, J.; Cumpston, M.; Li, T.; Page, M.J.; Welch V.A. (editors) Cochrane Handbook for Systematic Reviews of Interventions version 6.3 (updated February 2022) [internet]: Cochrane. 2022. Available online: https://training.cochrane.org/handbook/current
  8. Schardt, C.; Adams, M.B.; Owens, T.; Keitz, S.; Fontelo, P. Utilization of the PICO framework to improve searching PubMed for clinical questions. BMC Med Inform Decis Mak. 2007, 7, 16
  9. Balshem, H.; Helfand, M.; Schünemann, H.J.; Oxman, A.D.; Kunz, R.; Brozek Jet, et al. GRADE guidelines: 3. Rating the quality of evidence. J Clin Epidemiol. 2011, 64(4), 401-6
  10. Schünemann, H.; Brozek, J.; Guyatt, G.; Oxman, A.; editors. GRADE handbook for grading quality of evidence and strength of recommendations (updated October 2013) [internet]: The GRADE Working Group. 2013. Available online: https://gdt.gradepro.org/app/handbook/handbook.html
  11. Wells, G.A.; Shea, B.; O´Connell, D.; Petersen, J.; Welch Vet, et al. The Newcastle-Ottawa Scale (NOS) for assessing the quality of nonrandomized studies in meta-analyses. [internet]. Ottawa: Ottawa Hospital Research Institute. 2000. Available online: https://www.ohri.ca/programs/clinical_epidemiology/oxford.asp
  12. Lampmann, T.; Hadjiathanasiou, A.; Asoglu, H.; Wach, J.; Kern, T.; Vatter, H.; Güresir, E. Early Serum Creatinine Levels after Aneurysmal Subarachnoid Hemorrhage Predict Functional Neurological Outcome after 6 Months. J Clin Med. 2022, 11(16), 4753
  13. Jiang, R., Zhao, S., Wang, R., Feng, H., Zhang, J., Li, X., Mao, Y., Yuan, X., Fei, Z., Zhao, Y., Yu, X., Poon, WS., Zhu, X., Liu, N., Kang, D., Sun, T., Jiao, B., Liu, X., Yu, R., Zhang, J., Gao, G., Hao, J., Su, N., Yin, G., Zhu, X., Lu, Y., Wei, J., Hu, J., Hu, R., Li, J., Wang, D., Wei, H., Tian, Y., Lei, P., Dong, JF., Zhang, J. Safety and Efficacy of Atorvastatin for Chronic Subdural Hematoma in Chinese Patients: A Randomized Clinical Trial. JAMA Neurol.2018, 75(11), 1338-1346
  14. Hutchinson, P.J., Edlmann, E., Bulters, D., Zolnourian, A., Holton, P., Suttner, N., Agyemang, K., Thomson, S., Anderson, I.A., Al-Tamimi, YZ., Henderson, D., Whitfield, P.C., Gherle, M., Brennan, P.M., Allison, A., Thelin, E.P., Tarantino, S., Pantaleo, B., Caldwell, K., Davis-Wilkie, C., Mee, H., Warburton, E.A., Barton, G., Chari, A., Marcus, H.J., King, A.T., Belli, A., Myint, P.K., Wilkinson, I., Santarius, T., Turner, C., Bond, S., Kolias, A.G. British Neurosurgical Trainee Research Collaborative; Dex-CSDH Trial Collaborators. Trial of Dexamethasone for Chronic Subdural Hematoma. N Engl J Med. 2020, 383(27), 2616-2627
  15. Mebberson, K., Colditz, M., Marshman, L.A.G., Thomas, P.A.W., Mitchell, P.S., Robertson, K. Prospective randomized placebo-controlled double-blind clinical study of adjuvant dexamethasone with surgery for chronic subdural haematoma with post-operative subdural drainage: Interim analysis. J Clin Neurosci. 2020, 71, 153-157
  16. Ng, S., Boetto, J., Huguet, H., Roche, P.H., Fuentes, S., Lonjon, M., Litrico, S., Barbanel, A.M., Sabatier, P., Bauchet, L., Chevassus, H., Lonjon, N. HEMACORT Study Group. Corticosteroids as an Adjuvant Treatment to Surgery in Chronic Subdural Hematomas: A Multi-Center Double-Blind Randomized Placebo-Controlled Trial. J Neurotrauma. 2021, 38(11), 1484-1494

Round 2

Reviewer 1 Report

Thank you for your careful revision, especially the part about Method and Quality Assessment. Prospective studies in this paper are more meaningful than retrospective studies. However, considering that the limited studies were included, more caution is suggested in concluding.

The date (2015) is not replaced in Table 1. Please be consistent.

Author Response

Dear Reviewer

Thank you for reading our manuscript and critically reviewing it, which will help us improve it to a better scientific level and make it more understandable to the readership.

In the following we would like to respond to your remarks:

The reviewer is absolutely right that the results of the present meta-analysis are based on only five trials. Despite the high-quality (prospective, randomized, double-blind, and placebo-controlled) of the included trials, the results have to be interpreted with caution. We have implemented this potential limitation in the section “4.1 limitations”.

Furthermore, we have now revised and corrected the date of publication  (2015 –> 2016) of Prud´homme et al. [1] in the table 1. We apologize for not having revised this directly in the first round.

References

  1. Prud'homme, M., Mathieu, F., Marcotte, N., Cottin, S. A Pilot Placebo Controlled Randomized Trial of Dexamethasone for Chronic Subdural Hematoma. Can J Neurol Sci. 2016, 43(2), 284-90